# Dehydrin Protein TaCOR410 Improves Drought Resistance of Wheat Through Autophagy

**DOI:** 10.3390/plants14172726

**Published:** 2025-09-02

**Authors:** Mei Yan, Hua-Dong Song, Jia-Lian Wei, Kai-Yong Fu, Gang Li, Yong-Bo Li, Cheng Li

**Affiliations:** 1Agricultural College, Shi He Zi University, Shihezi 832000, China; yanmei1029@163.com (M.Y.); m18386191910@163.com (J.-L.W.); ligang@xjshzu.com (G.L.); 2Shandong Luyan Seed Co., Ltd., Jinan 250100, China; shd9229@sina.com; 3National Key Laboratory of Wheat Improvement, Crop Research Institute, Shandong Academy of Agricultural Sciences, Jinan 250100, China

**Keywords:** drought, wheat, autophagy, TaATG8, TaCOR410

## Abstract

Drought seriously affects wheat yield; it is therefore important to study the molecular mechanism of wheat resistance to drought stress to ensure national food security. Plants can remove harmful substances through autophagy, thus improving their drought resistance. The results of previous studies have shown that autophagy is involved in the drought stress response; however, the molecular mechanism of autophagy in response to drought stress has yet to be elucidated. In this study, molecular biological methods such as immunohistochemistry, Co-Immunoprecipitation (Co-IP), and pull-down were used to explain the molecular mechanism of autophagy in response to drought stress at the protein level. We found that a dehydrin protein called cold-regulated 410 (TaCOR410) interacts with autophagy-related 8 (TaATG8, a key protein of wheat autophagy). TaCOR410 interacted with TaATG8 through its ATG8-interacting motif (AIM), and interaction was inhibited after mutation of the AIM. Interference with *TaCOR410* inhibited autophagy and reduced the drought resistance of wheat. In contrast, transient transfection of *TaCOR410* promoted autophagy. In wheat, overexpression of *TaATG8* improved the drought resistance of wheat. Following interference with *TaATG5*, *TaATG7* inhibited autophagy and reduced the drought resistance of wheat. From the above results, it is evident that autophagy can improve the drought resistance of wheat and can respond to drought stress through the interaction of TaCOR410 with TaATG8.

## 1. Introduction

By 2025, it will be necessary for crop yields in drought-prone areas to be increased by 40% to feed the ever-increasing human population [1]. Global losses in crop production due to drought have amounted to approximately 30 billion US dollars over the past decade [2]. Bread wheat (*Triticum aestivum* L.) provides roughly 19% of global dietary energy [3]. Drought is a major environmental factor that limits wheat production worldwide [4]. In recent years, climate change has further exacerbated the problem and resulted in a higher frequency of extreme weather changes, leading to abiotic stress conditions [5]. To adapt to stress conditions, plant cells must recycle damaged or unwanted proteins and organelles by means of autophagy, which is a conserved eukaryotic mechanism employed for the degradation of cellular components in lytic organelles (vacuoles in yeast and plants and lysosomes in animals) [6]. Upon induction of autophagy, a double-membrane structure engulfs unwanted cellular components to form an autophagosome and transports them to the central vacuole for degradation and recycling [7]. A number of autophagy-related (ATG) proteins have been identified for autophagosome formation and delivery to vacuoles [8,9]. Autophagy can regulate cellular homeostasis by selectively degrading specific cargo materials [10]. The specificity of the cargo is determined by selective autophagy receptors, which function as sorting adaptors that recruit selected cargo into autophagosomes through their ability to interact with ATG8 proteins [11]. ATG8 is a ubiquitin-like protein modified by phosphatidylethanolamine (PE) for incorporation into the autophagosome membrane [8]. Specific interactions between selective autophagy receptors and ATG8 require the ATG8-interacting motif (AIM), a short motif within the selective autophagy receptor [12].

Autophagy plays an important role in plant drought resistance. In *Arabidopsis thaliana ATG5* and *ATG7* mutants, autophagy is blocked, and their drought resistance is reduced [13,14]. Tomato is hypersensitive to drought after interference with *ATG8d* and *ATG18h* [15]. Overexpression of *MdATG8i* or *MdATG18a* enhanced drought tolerance in transgenic apple [16,17]. In our previous study, drought induced the expression of wheat *ATGs*, and an increase in autophagy puncta and ATG8-PE; in addition, interference of *ATG6* led to a reduction in drought tolerance [18]. However, the molecular mechanism of autophagy in response to drought stress in wheat requires further analysis.

Late embryogenesis abundant (LEA) proteins are an important group of functional proteins that reduce cell damage and protect cells under abiotic stress conditions [19]. The results of previous studies have suggested that dehydrins (DHNs), also known as LEA II proteins, exhibit various types of biochemical activities, such as ion isolation, membrane stabilization, and chaperoning [20]. DHNs accumulate during late seed developmental stages and in vegetative tissues subjected to dehydration, salt, and low-temperature stresses [21]. In general, DHNs contain three highly conserved K-, Y-, and S-segments: the K-segment is rich in lysine (EKKGIMDKIKEKLPG) and very conserved among different species; it is located near the C-terminal and can form an α-helix with both hydrophilic and hydrophobic characteristics, which is related to the combination of macromolecules [22]. The Y-segment (T/VDEYGNP) is partially similar to the nucleotide-binding site of the molecular chaperone, and it is usually located at the N-terminal of DHN [23]. The S-segment is composed of 4–10 serine residues, and it is the main site of phosphorylation [24]. Based on the number and arrangement characteristics of conserved sequences, DHNs are divided into five subgroups: YnSKn, YnKn, SKn, Kn, and KnS. Different subgroups of DHNs respond to different environmental stresses, and the YnSKn subgroup is mainly induced by dehydration stress and abscisic acid (ABA). The Kn and YnKn subgroups are mainly activated by low-temperature stress, the SKn subgroup responds to drought, low temperature, and salt stresses, and the KnS subgroup responds to drought and low-temperature stresses [25]. DHNs impart drought stress tolerance by enhancing water retention capacity, increasing chlorophyll content, maintaining photosynthetic machinery, activating reactive oxygen species (ROS) detoxification, and promoting the accumulation of compatible solutes [26]. The interaction of rice YSK2-type dehydrin (OsDhn-Rab16D) with a prolyl cis-transisomerase (OsFKBP, Os02g52290) improved drought resistance through ABA signaling [27]. The DHN MtCAS31 (cold acclimation-specific 31) promotes autophagic degradation under drought stress in *Medicago truncatula* [28]. Fifty-five DHN genes have been found in wheat [29]. Wheat TaCOR410 is a DHN that accumulates around the plasma membrane, is present in lesser amounts in the intercellular space [30], and is upregulated under drought stress [31].

The results of previous studies have shown that autophagy is involved in the drought stress response; however, the molecular mechanism of autophagy in response to drought stress has yet to be elucidated. In our study, TaCOR410 interacted with TaATG8 to promote wheat autophagy. Overexpression of *TaATG8* improved the drought resistance of wheat, whereas inhibition of autophagy decreased its drought resistance.

## 2. Results

### 2.1. Drought Induces Autophagy in Wheat

To investigate whether autophagy can respond to drought stress, ATG8-labeled autophagy puncta were used to detect plant tissue autophagy [32], and the expression of key autophagy genes was determined under dehydration conditions. The results showed that the abundance of autophagy puncta increased significantly under dehydration for 3–24 h when compared with the control (H_2_O); with the increase in dehydration time, the greater the abundance of autophagy puncta (Figure 1A,B). Additionally, our qRT-PCR results showed that the expression of key autophagy genes *ATG3*, *ATG4*, *ATG5*, *ATG6*, and *ATG7* was significantly upregulated within 3–24 h (Appendix A). These findings provide conclusive evidence that drought induces autophagy in wheat.

### 2.2. Screening and Identification of TaATG8 Interaction Protein TaCOR410

To investigate the molecular mechanism underlying the autophagy response to drought stress, a library of ATG8 interaction proteins was established by immunoprecipitating the total proteins of wheat leaves with a wheat ATG8 antibody and performing a mass spectrometry analysis. In this library, a wheat DHN protein, TaCOR410, was identified (Figure 2A). Due to DHNs being intrinsically disordered proteins and showing a tendency to interact with other biomolecules, we investigated whether TaCOR410 functions as a cargo receptor in the autophagy pathway. In general, ATG8 can bind to its interactors through the ATG8 interaction motif (AIM; W/F/YX1X2L/I/V) [33,34]. Hence, we analyzed the sequence of TaCOR410 and the predicted 1 AIM (75-FSKL-78). In addition, the protein contains 262 amino acids, an S-segment (81–89 amino acids), and two K-segments (175–189 and 227–241 amino acids) (Figure 2B), with it belonging to SK2 DHN.

### 2.3. TaCOR410 Specifically Interacts with TaATG8

To further confirm the interaction of TaCOR410 with TaATG8, bimolecular fluorescence complementation (BiFC), co-immunoprecipitation (Co-IP), and pull-down experiments were conducted. The BiFC results showed that the yellow fluorescence signal was observed only when TaATG8-nYFP and TaCOR410-cYFP were co-transfected into the tobacco leaves and wheat protoplasts; in comparison, no yellow fluorescence was detected in the negative controls. However, the yellow fluorescence signal disappeared when the AIM FSKL (TaCOR410-cYFP) was mutated into ASKA (TaCOR410^ASKA^-cYFP) (Figure 3A–C).

In addition, Co-IP was also performed to verify the results. TaCOR410-flag/GFP-TaATG8 or TaCOR410^ASKA^-flag/GFP-TaATG8 were co-expressed in tobacco leaves. Protein was immunoprecipitated with anti-GFP beads and detected with anti-flag. Only TaCOR410-flag and GFP-TaATG8 were able to bind specifically (Figure 4A). Lastly, we performed pull-down assays to confirm this interaction using GFP, TaCOR410-GFP, or TaCOR410^ASKA^-GFP protein as bait to precipitate the His-TaATG8 protein. As shown in Figure 4B, the TaCOR410-GFP protein bound specifically to His-TaATG8; however, neither GFP nor TaCOR410^ASKA^-GFP exhibited the same behavior. From the above results, it can be concluded that wheat DHN protein TaCOR410 interacts with TaATG8 in the AIM both in vitro and in vivo.

### 2.4. Interference with TaCOR410 Reduces the Drought Resistance of Wheat

To investigate the role of *TaCOR410* in wheat drought resistance, the drought resistance phenotype was identified after *TaCOR410* was silenced by using the barley stripe mosaic virus (BSMV)-based virus-induced gene silencing (VIGS) method [35]. After *TaCOR410* interference (*TaCOR410i*), the seedlings exhibited obvious dryness, and their fresh weight showed evident reduction when compared with the control (*GFPi*) after 25 days of drought stress (Figure 5A,B,D). The qRT-PCR results showed that *TaCOR410* was significantly upregulated under drought stress from 6 to 48 h (Figure 5C). The results suggest that interference with *TaCOR410* reduces the drought resistance of wheat, indicating that *TaCOR410* responds to drought stress.

### 2.5. TaCOR410 Promotes Autophagy in Wheat

To examine whether TaCOR410 plays a role in drought-induced autophagy, we investigated the effects of *TaCOR410* on autophagy under drought stress. The immunohistochemical results showed that interference with *TaCOR410* led to a significant decrease in autophagy puncta under drought stress (Figure 6A,B). In addition, GFP-TaATG8 labeled autophagy puncta and cleavage of GFP-TaATG8 were measured to indicate autophagy after transfection of *TaCOR410*. In tobacco, co-transfection of *TaCOR410-flag*/*GFP-TaATG8* produced more significant autophagy puncta than that of *flag/GFP-TaATG8* (Figure 6C,D). Overexpression of *TaCOR410-flag* released more free GFP than flag (Figure 6E). These findings demonstrate that interference with *TaCOR410* inhibits the formation of autophagy puncta and overexpression of *TaCOR410* promotes the formation of autophagy puncta and cleavage of GFP-TaATG8. From the above results, it can be concluded that *TaCOR410* promotes autophagy in wheat.

### 2.6. Inhibition of Autophagy Reduces the Drought Resistance of Wheat

To elucidate the effects of autophagy on the drought resistance of wheat, the drought resistance phenotype of the seedlings was detected after autophagy was blocked by interference with *TaATG5* or *TaATG7*. *ATG5* and *ATG7* play key roles in autophagy, and inhibition of *ATG5* or *ATG7* represses the autophagy pathway [36].

Our results demonstrated that interference with *TaATG5* or *TaATG7* led to a significant reduction in autophagy puncta, and seedlings wilted faster and the fresh weight of leaves decreased significantly under drought stress (Figure 7A–E), which indicates that inhibition of autophagy will accelerate the premature senescence of seedlings caused by drought.

### 2.7. Overexpression of TaATG8 Improves the Drought Resistance of Wheat

To investigate the role of *TaATG8* under drought stress, we generated transgenic plants overexpressing *TaATG8* (OE1, OE2, OE3), driven by the maize *ubiquitin* promoter, and Fielder was used as the receptor material. The transgenic lines wilted relatively slowly after 10 days of drought stress (Figure 8A). After 3 days of rehydration, the transgenic lines were reddish green; in comparison, the control Fielder were still dry (Figure 8B). In addition, the results of the drought simulation experiment showed that the root and bud lengths of the transgenic lines were significantly greater than those of recipient wheat Fielder (Figure 8C,D). All *TaATG8* transgenic lines showed positive signals (Figure 8E), and the expression of *TaATG8* was significantly higher than that of the receptor (Figure 8F). These results demonstrate that overexpression of *TaATG8* can improve the drought resistance of wheat at the seedling stage.

## 3. Discussion

Autophagy is essentially a process of digesting, degrading, and recycling harmful components in cells, and it can help plants survive under drought stress [37,38]. The results of a previous study demonstrated that drought promotes the expression of *ATGs* and the formation of ATG8-PE, thus activating autophagy in wheat [18]. However, the molecular mechanism of autophagy in response to drought has yet to be elucidated in wheat. In this study, we found that autophagy can respond to drought stress through the interaction of TaATG8 with TaCOR410. Interference with *TaCOR410* inhibits autophagy and reduces the drought resistance of wheat, and transient transfection of *TaCOR410* promotes autophagy. TaCOR410 interacts with TaATG8 through its AIM region, thus promoting autophagy. Overexpression of *TaATG8* improves the drought resistance of wheat, and inhibition of autophagy reduces the drought resistance of wheat.

### 3.1. TaCOR410 Improves the Drought Resistance of Wheat Seedlings by Regulating Autophagy

DHNs are molecular protectors that protect membranes and macromolecules from denaturation under abiotic stress [39]. In wheat, TaCOR410 is a DHN that accumulates around the plasma membrane, and it is present in lesser amounts in the intercellular space [30]. In wheat, DHNs have been reported to accumulate in response to drought stress during anthesis [40]. The results of a previous study demonstrated that the number of mRNA transcripts of *TaCOR410* significantly increased as water availability decreased in all wheat cultivars during the post-germination stage, presumably to enhance plant tolerance to drought stress through cell membrane protection, cryoprotection of enzymes, and prevention of ROS accumulation [41]. However, the molecular mechanism of TaCOR410 enhancing wheat drought stress requires further elucidation. Our findings demonstrated that interference with *TaCOR410* inhibits drought-induced autophagy, resulting in decreased drought resistance in wheat. In contrast, overexpression of *TaCOR410* promotes autophagy. Inhibition of autophagy leads to a decrease in drought resistance. From the above results, it can be concluded that TaCOR410 improves the drought resistance of wheat seedlings by regulating autophagy.

### 3.2. Autophagy Responds to Drought Stress Through the Interaction of TaCOR410 with TaATG8

Post-translational regulation is an important method for autophagy to cope with drought stress, and protein interaction is a form of post-translational regulation that exerts its effects in adaptation to various environmental stresses by promoting or inhibiting autophagy [42]. Cargo receptors play important roles in conferring autophagy selectivity by interacting with ATG8 via AIMs or the ubiquitin-interacting motifs [12,43]. To date, some cargo receptors related to plant drought resistance have been identified. In *M. truncatula*, the DHN MtCAS31 acts as a cargo receptor to form the plasma membrane intrinsic protein 2 (MtPIP2;7)-cold acclimation-specific 31 (MtCAS31)–MtATG8 complex, which facilitates MtPIP2;7 autophagic degradation and thus reduces water loss under drought stress and improves drought tolerance [28]. In *Arabidopsis*, the cargo receptor tryptophan-rich sensory protein/translocator (TSPO) interacts with ATG8 to promote the autophagic degradation of PIP2, thus improving drought resistance [44]. DOMINANT SUPPRESSOR OF KAR 2 (DSK2) is a ubiquitin-binding cargo receptor involved in the autophagic degradation of BRI1-EMSSUPPRESSOR 1 (BES1) to improve drought resistance mediated by the BR signaling pathway [45,46]. In this study, a cargo receptor, TaCOR410, was screened and identified from the interaction protein library of TaATG8; TaCOR410 can promote autophagy through the interaction of the AIM with TaATG8, thus improving the drought resistance of wheat. It cannot specifically bind to TaATG8 if the AIM is mutated. However, whether the interaction of TaCOR410 with TaATG8 is required to promote the autophagic degradation of PIP2 or any other substance in wheat requires further elucidation.

### 3.3. ATG8 Plays a Crucial Role in Wheat Drought Resistance

*ATG8* plays an important role in maintaining the normal growth and development of plants and coping with various biotic and abiotic stresses. ATG8 is conjugated to the membrane lipid PE in a ubiquitin-like conjugation reaction that is essential for autophagosome formation [47]. In one study, *Gossypium* GhATG8f improved the salt tolerance of cotton by increasing superoxide dismutase, peroxidase, and catalase activities and proline accumulation [48]. Decreased *ATG8F* expression repressed the salt tolerance of transgenic rapeseed plants by significantly reducing root Na(+) retention under salt stress conditions and decreasing N uptake and translocation under N starvation [49]. ATG8b-mediated autophagy is involved in N recycling to grains and contributes to the grain quality of rice [50]. Silencing of *ATG8c* in the Chinese white pear (*Pyrus bretschneideri Rehder*) decreased its resistance to *Botryosphaeria dothidea* [51]. ATG8 is a positive regulator of the osmotic and drought stress response in wild emmer wheat [52]. In bread wheat, *ATG8* is involved in grain development; interference with *ATG8* will inhibit pericarp degradation, and the grain will become smaller and premature [53]. TaATG8j promotes resistance by regulating cell death in wheat affected by stripe rust [54]. In wheat, the function of *TaATG8* has been investigated by means of only instantaneous interference. In this study, the role of *TaATG8* in drought resistance was investigated using wheat transgenic materials with stable inheritance. In wheat, overexpression of *TaATG8* can significantly increase the length of wheat seedling roots and buds and increase their resistance to drought stress.

## 4. Materials and Methods

### 4.1. Plant Material

The wheat cultivar Fielder was used as the receptor material in this study. Jimai 379, cultivated at the Crop Research Institute, Shandong Academy of Agricultural Sciences, is a water-saving and drought-resistant wheat variety.

### 4.2. Wheat Transformation

*TaATG8* transgenic materials were obtained by Professor Li Genying’s group from the Crop Research Institute, Shandong Academy of Agricultural Sciences. Briefly, the open reading frame (ORF) of *TaATG8* (*TraesCS2A02G224000*) was cloned into the binary vector *pLGY02* at the SmaI and SpeI sites, forming a *pLGY02-TaATG8* recombinant plasmid and maize *Ubiquitin* promoter-driven target gene expression. The construct *pLGY02-TaATG8* was used to transform the drought-sensitive variety Fielder through *Agrobacterium tumefaciens* (*Agrobacterium*)-mediated transformation [55]. T1–T4 plants were tested for the presence of the transgene via PCR amplification, and Fielder was used as the negative control. Phenotypic identification of three group plants from each T4 line was performed at the seedling stage.

### 4.3. Wheat Seedlings Subjected to Drought Stress Treatment

Seeds of the wheat cultivar Jimai 379 were germinated at 22 °C. After 1 week, seedlings that showed consistent growth were selected for analysis. These seedlings were incubated in Hoagland nutrient solution for 3 days. Thereafter, some seedlings were transplanted into 20% PEG-8000 solution (P8260; Solarbio Science & Technology Co., Ltd., Beijing, China) to facilitate dehydration. Other seedlings were transplanted into flowerpots filled with peat soil soaked in Hoagland nutrient solution for drought stress experiments. The treatments were performed in a controlled environment chamber under a 16:8 h light–dark photoperiod and 22 °C temperature conditions.

### 4.4. Quantitative Real-Time Reverse Transcription (qRT)-PCR

Total RNA was extracted from wheat leaves by using a Rapid RNA extraction kit for plant tissues (DP452; TIANGEN, Beijing, China). After determining RNA quality via electrophoresis on a 1% agarose gel, 5 μg of RNA was reverse-transcribed into cDNA by using TransScript^®^ First-Strand cDNA Synthesis SuperMix (AT301-02; TransGen Biotech, Beijing, China). The cDNAs were used as templates for qRT-PCR performed using TransStart^®^ Top Green qPCR SuperMix (AQ131-01; TransGen Biotech), according to the manufacturer’s instructions; β-actin was amplified for internal standardization. The experiments were repeated three times, and the experimental data were analyzed using Student’s *t*-test. The relative expression data were analyzed using the 2^−ΔΔCT^ method from the qRT-PCR experiments. The primers used are listed in Appendix A.

### 4.5. Preparation of the TaATG8 Polyclonal Antibody

For the preparation of the TaATG8 polyclonal antibody, we followed the method outlined in a previous study [56]. Briefly, one Japanese white rabbit and one New Zealand white rabbit of experimental grade were raised. When the rabbits grew to 1–2 kg, 1 mL of well-mixed complete Freund’s adjuvant and 1 mL of TaATG8 polypeptide (N-KTEHPLERRQAESARIRE-C) at a concentration of 0.7 mg/mL were injected subcutaneously into each rabbit with a syringe. This stage was marked as day 1. On day 12, 1 mL of well-mixed incomplete Freund’s adjuvant and 1 mL of polypeptide at a concentration of 0.35 mg/mL were injected subcutaneously into the rabbits. On days 26, 40, and 54, 1 mL of well-mixed incomplete Freund’s adjuvant and 1 mL of polypeptide at a concentration of 0.35 mg/mL were injected intramuscularly into the rabbits. On day 66, the rabbit serum was taken for Western blot verification.

### 4.6. Establishment of the TaATG8 Interacting Protein Library

First, 1 mL of total protein of wheat leaf at a concentration of 2 mg/mL and 5 μL TaATG8 antibody at a concentration of 1.8 mg/mL were mixed and incubated on ice overnight; thereafter, 50 μL of Protein G Agarose (P2053, Beyotime, Shanghai, China) was added and incubated on ice for 3 h; it was then centrifuged at 400× *g* for 5 min and the supernatant was discarded. The gel-like substance at the bottom of the tube was retained, washed 5 times with PBS, and 40 μL of PBS was added and mixed after homogenization. Next, 20 μL of 5× protein loading buffer (G2075, Servicebio, Wuhan, China) was added, boiled for 10 min, and then proceeded by means of SDS-PAGE electrophoresis, with subsequent staining with SDS gel. The gel was subjected to mass spectrometry identification and analysis after decolorization.

The samples were processed using a standard in-gel digestion protocol, with trypsin used for in-gel digestion of the gel slices [57]. The specific experimental steps were as follows: the stained gel was incubated with trypsin (Promega) overnight at 37 °C. Peptides were extracted by means of incubation with 5% tallow fatty acid (TFA) for 1 h, followed by the addition of 2.5% TFA and 50% acetonitrile (ACN) at 37 °C for 1 h. The cleaved peptides were analyzed using a Nano LC-LTQ Orbitrap XL Liquid Chromatography–tandem mass spectrometry (LC-MS/MS) System (Thermo Scientific, Waltham, MA, USA). Data were analyzed using pFindStudio software (v.3.1.6) [58].

### 4.7. Western Blotting

Total proteins from wheat or tobacco leaves were extracted using the Plant Protein Extraction Kit (CW0885M; CoWin Biotech Co., Ltd., Taizhou, China). Equal amounts of protein (20 μg) from each sample were subjected to SDS-PAGE. The proteins were transferred electrophoretically onto a nitrocellulose membrane. The membrane was incubated with blocking buffer (2% skim milk powder dissolved in TBS) for 1 h at room temperature. Next, the membrane was incubated with a rabbit anti-ATG8 polyclonal antibody, rabbit anti GFP-Tag polyclonal antibody (AE011; ABclonal, Wuhan, China), or Anti-Rubisco mouse monoclonal antibody (Jining Bio, Shanghai, China) diluted 1:5000 in blocking buffer in TBS and incubated with the membrane for 2 h at room temperature. Thereafter, the membrane was washed with TBST (2% TWEEN-20 dissolved in TBS) 3 times and incubated with alkaline phosphatase-conjugated goat anti-rabbit IgG (A0239; Beyotime, Shanghai, China) diluted 1:10000 in the blocking buffer for 2–3 h at room temperature. The protein signal was visualized using the BCIP/NBT alkaline phosphatase chromogenic kit (C3266; Beyotime), according to the manufacturer’s instructions. The Prestained Protein Marker II (G2058, Servicebio, Wuhan, China) was used as a protein ruler.

### 4.8. Knock Down of TaATG5, TaATG7, and TaCOR410

A 206 bp fragment of *TaATG5*, a 216 bp fragment of *TaATG7*, and a 235 bp fragment of *TaCOR410* from the conserved coding sequence were amplified and purified, and a 300 bp fragment of GFP was used as the control. The γ strand of BSMV was digested using *ApaI* and fused with the *TaATG5*, *TaATG7*, *TaCOR410*, and GFP fragments to form the vectors BSMVγ-*TaATG5*, BSMVγ-*TaATG7*, BSMVγ-*TaCOR410*, and BSMVγ-GFP, respectively. BSMVα, BSMVβ, BSMVγ-*GFP*, BSMVγ-*TaATG5*, BSMVγ-*TaATG7*, and BSMVγ-*TaCOR410* were transformed into *Agrobacterium* GV3101. *Agrobacterium* GV3101 containing the target plasmid was cultured in YEB medium until OD = 0.6–0.8. The bacterial culture containing α, β, and γ components was mixed at a ratio of 1:1:1 and maintained at 28 °C for 4 h. Unfolded tobacco leaves at the 4–5 leaf stage were injected with the *Agrobacterium* solution. After 7–12 days, the injected leaves were collected and fully ground in buffer (1 g wheat leaf in 2–3 mL PBS buffer and 1% diatomite; pH = 7.2), and 15 µL of the juice was rubbed and inoculated onto the second leaf of wheat at the two-leaf and one-core stage. Six to eight days after inoculation, the target gene was detected using qRT-PCR.

### 4.9. Immunohistochemistry

Place the first leaf of the treated wheat seedling on the operating table, and gently make an incision in the middle part of the leaf with the tip of a dissecting knife. Then, using forceps, carefully peel off the lower epidermis from the leaf along the incision. The lower epidermis was treated with 4% paraformaldehyde at 4 °C overnight, placed in BSA and sealed at 37 °C for 1 h, incubated with a rabbit anti-ATG8 antibody diluted 1:200 in blocking buffer in TBS and incubated at 4 °C overnight, washed 3 times with PBS, incubated with the secondary antibody AF488-labeled Goat Anti-Rabbit IgG (A0423, Beyotime, Shanghai, China) diluted 1:1000 in BSA at 37 °C for 1 h, and washed 3 times with PBS. Thereafter, the lower epidermis tissue of the wheat leaves was placed on a glass slide for tableting, and a fluorescence microscope (HT7700; Hitachi, Tokyo, Japan) and a confocal microscope (FV3000; Olympus, Tokyo, Japan) were used to observe the samples and obtain images.

### 4.10. Observing Autophagy Puncta in Tobacco

*GFP* was linked with the N-terminal ORF of *TaATG8* to generate a *GFP-TaATG8* fragment. The ligated fragment was cloned into a *35S-pBWA(V)HS-CCDB* vector at BsaI and Eco31I sites, forming a *35S-GFP-TaATG8* reporter plasmid, which was constructed and transferred into *Agrobacterium* GV3101. When the OD of the bacterial culture reached 0.6–0.8, the culture was injected into tobacco with 4–6 leaves. Additionally, 100 μM E64D was injected into tobacco to inhibit enzyme activity in vacuoles. Autophagy puncta were observed using the aforementioned confocal microscope 48 h after injection.

### 4.11. GFP Cleavage Assay

The ORF of *TaCOR410* (TraesCS6D02G234700) was cloned into the *35S-pBWA(V)KS-CCDB–flag* vector at NdeI and EcoRI sites, forming a *35S-TaCOR410-Flag* recombinant plasmid. *GFP* was linked with the N-terminal ORF of *TaATG8* to generate the *GFP-TaATG8* fragment; thereafter, the ligated fragment was cloned into the *35S-pBWA(V)HS-CCDB* vector at the BsaI and Eco31I sites, forming a *35S-GFP-TaATG8* recombinant plasmid. The above plasmids, *35S-GFP-TaATG8*, *35S-TaCOR410-flag*, and *35S-flag*, were transferred into *Agrobacterium* GV3101. The bacterial culture containing *35s-GFP-TaATG8* and the bacterial culture containing *35S-TaCOR410-flag* or *35S-flag* were evenly mixed at a ratio of 1:1 and centrifuged, and the supernatant was discarded; tobacco infection liquid was added to resuspend the bacteria and then injected into tobacco leaves, which were cultured for 48 h. The protein extract was analyzed with anti-GFP, and GFP-TaATG8 and free GFP were detected. In this assay, Rubisco was used as the reference.

### 4.12. BiFC Assay

The *TaATG8* coding sequence was fused with the *pCAMBIA1300-35S-C-YFPN* vector at the BamHI and XbaI sites to generate the *TaATG8-nYFP* plasmid. In addition, *TaCOR410* and *TaCOR410^ASKA^* were fused with the *pCAMBIA1300-35S-C-YFPC* vector at the BamHI and XbaI sites to generate *35S-TaCOR410-cYFP* and *35S-TaCOR410^ASKA^-cYFP* plasmids, which were separately transferred into *Agrobacterium* GV3101, and infiltration of tobacco was performed using a previously described method [59]. Infected tissues were analyzed at 48 h after infiltration. Fluorescence staining was visualized using a confocal microscope.

### 4.13. Pull-Down Assay

To investigate the interaction between TaCOR410 and TaATG8, the full-length coding sequence of *TaATG8* was subcloned into the *pET-30a* vector and expressed in BL21 cells to produce a His-TaATG8 fusion protein. The *35S-TaCOR410-GFP*, *35S-TaCOR410^ASKA-^GFP*, and *35S-GFP* plasmids were transferred into *Agrobacterium* GV3101 and injected into tobacco leaves. The total protein in tobacco was extracted after 48 h of culture. Different target proteins were incubated with 1 μg His-TaATG8 at 4 °C for 1 h with shaking. Thereafter, the GFP antibody was added, followed by shaking at 4 °C overnight. Next, protein A agarose was added to the samples for 2–3 h at 4 °C. The samples were centrifuged, and the precipitate was collected and washed with PBS 5 times. The precipitate was boiled in 2× SDS loading buffer, and the proteins were analyzed through immunoblotting using the anti-His tag (AE068; ABclonal) and anti-GFP tag.

### 4.14. Co-IP

*Agrobacterium* GV3101 harboring *35S-GFP*/*35S-TaCOR410-flag*, *35S-GFP-TaATG8* /*35S-TaCOR410-flag*, or *35S-GFP-TaATG8*/*35S-TaCOR410^ASKA^-flag* constructs was injected into tobacco leaves. The total protein in tobacco leaves was extracted after 48 h of culture; 5 μL anti-GFP tag antibody was added to 2 mL protein extract and incubated overnight at 4 °C. The extract was centrifuged, and the precipitate was collected and washed with PBS 5 times. The precipitate was boiled in 2× SDS loading buffer. For immunoblotting analysis, anti-GFP and anti-FLAG (AF0036, Beyotime) antibodies were used to detect the relevant fusion proteins.

### 4.15. Quantitative Statistics

Quantitative analysis of autophagy puncta using ImageJ1.8.0 software. The data are presented as the mean ± SD. Significant differences between the two groups were determined using Student’s *t*-test with GraphPad Prism 5 software. Asterisks indicate statistically significant differences (*, *p* < 0.05; **, *p* < 0.01).

## 5. Conclusions

Based on the results of this study, we have produced a model to explain the molecular mechanism of autophagy in response to drought stress. Autophagy is involved in the drought stress response. TaCOR410 interacts with TaATG8 through its AIM to promote autophagy. Interference with *TaCOR410* represses autophagy, and inhibition of autophagy reduces the drought resistance of wheat; in comparison, overexpression of *TaATG8* improves the drought resistance of wheat. Overall, drought stress promotes autophagy through the interaction of TaCOR410 with TaATG8 (Figure 9).

## Figures and Tables

**Figure 1 plants-14-02726-f001:**
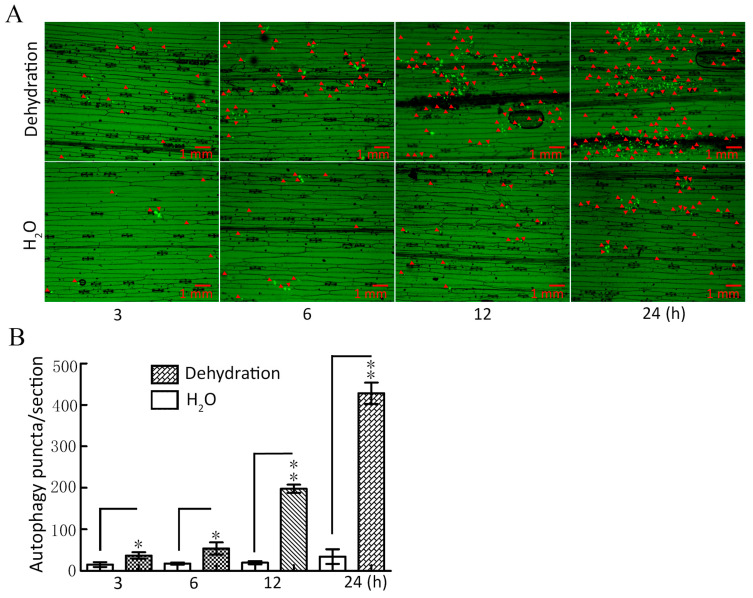
Assay for autophagy puncta under dehydration conditions. (**A**) TaATG8-labeled autophagy puncta was detected under dehydration. Dehydration: 20% PEG-8000 was used to simulate drought stress at the indicated times. H_2_O was used as the control. The red arrows indicate autophagy puncta. Scale bars = 1 mm. (**B**) Significance analysis of the number of autophagy puncta per section in (**A**). Three sections were used for analysis. All experiments were conducted in triplicate. The error is the average SD of three experiments, and significant differences between the experimental and control groups were analyzed using the *t*-test (* *p* < 0.05; ** *p* < 0.01).

**Figure 2 plants-14-02726-f002:**
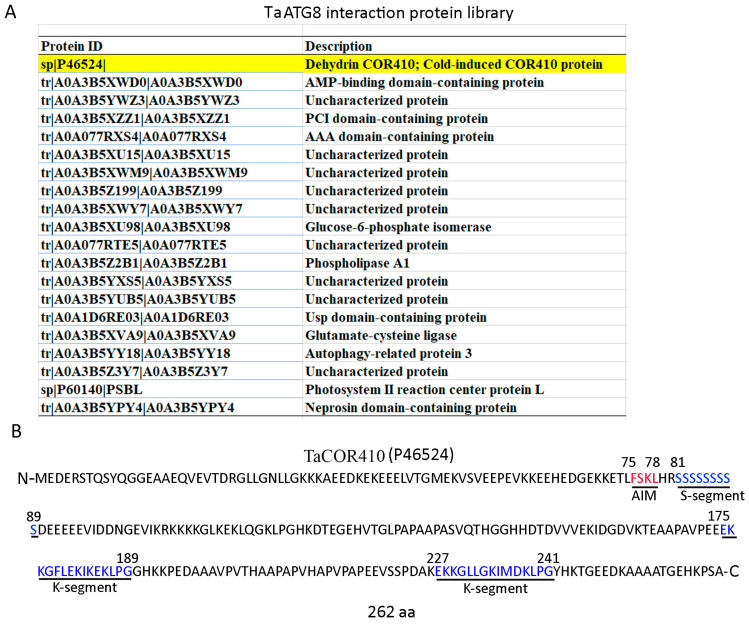
Screening and identification of TaCOR410. (**A**) The TaATG8 interactive protein library; the screened TaCOR410 protein is highlighted in yellow. (**B**) TaCOR410 protein sequence analysis. The red letters represent the AIM, and the blue letters represent the S-segment and two K-segments. The numbers in the sequence represent the positions of amino acids.

**Figure 3 plants-14-02726-f003:**
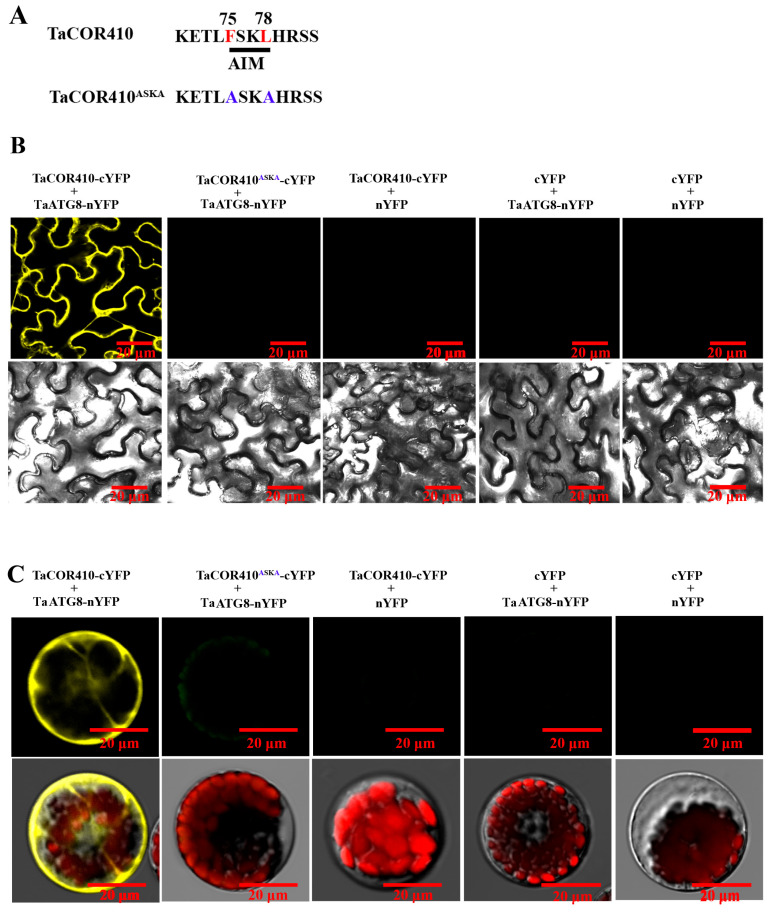
BiFC assay confirming the interaction of TaCOR410 with TaATG8 in vivo. (**A**) TaCOR410 was mutated into TaCOR410^ASKA^. The 75–78th sequence is the AIM. The red letters indicate wild-type sites. The blue letters indicate mutant sites. (**B**) The BiFC assay was conducted using tobacco. TaATG8-nYFP and TaCOR410-cYFP are experimental groups, and the other groups are negative controls. (**C**) The BiFC assay was performed using the wheat protoplast. Images of all cells were obtained using a confocal microscope. Scale bars = 20 μm. All experiments were conducted in triplicate.

**Figure 4 plants-14-02726-f004:**
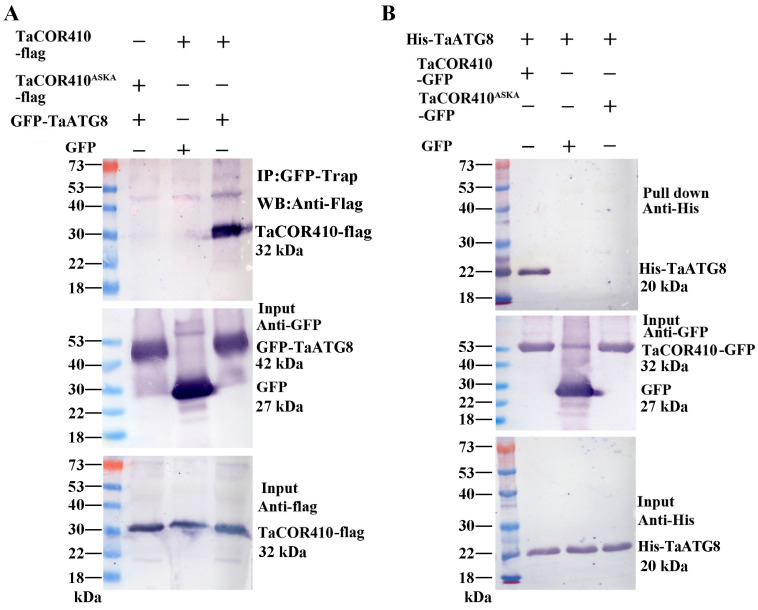
Co-IP and pull-down assays to verify the interaction between TaATG8 and TaCOR410. (**A**) Detection of TaCOR410-TaATG8 interaction via Co-IP. TaCOR410 and TaCOR410^ASKA^ were tagged with flag (TaCOR410-flag and TaCOR410^ASKA^-flag). GFP was tagged with TaATG8 (GFP-TaATG8). TaCOR410-flag/GFP and GFP-TaATG8/TaCOR410^ASKA^-flag were used as negative controls. Total proteins were extracted from tobacco leaves co-transformed with the indicated constructs and incubated with GFP beads to immunoprecipitate the target protein. Coprecipitated proteins were analyzed with Western blotting by using anti-flag and anti-GFP antibodies. (**B**) Detection of TaCOR410-TaATG8 interaction using the pull-down assay. His tag was fused with TaATG8 (His-TaATG8), and TaCOR410 and TaCOR410^ASKA^ were fused with a GFP tag (TaCOR410-GFP and TaCOR410^ASKA^-GFP). TaCOR410-GFP, GFP, or TaCOR410^ASKA^-GFP alone was incubated with His-TaATG8 and precipitated with GFP beads. The precipitates were separated using SDS-PAGE and analyzed with Western blotting by using anti-His and anti-GFP antibodies. All experiments were conducted in triplicate.

**Figure 5 plants-14-02726-f005:**
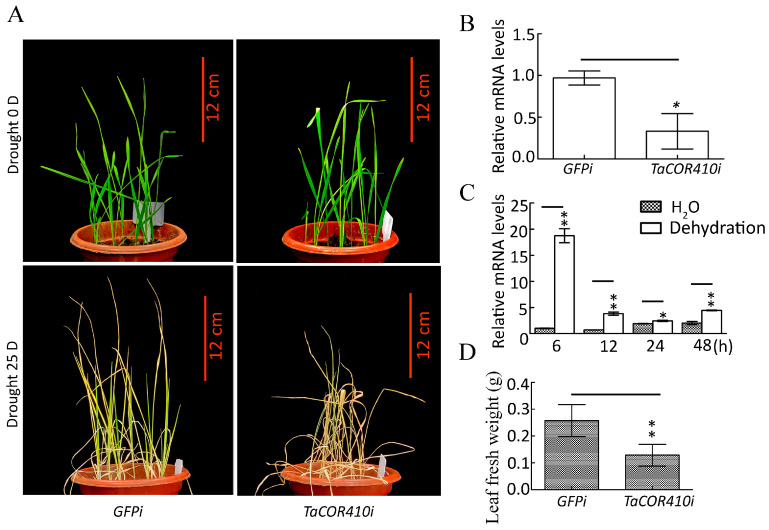
Phenotype of wheat seedlings after *TaCOR410* interference and expression of *TaCOR410* under dehydration treatment. (**A**) Phenotype of wheat seedling after *TaCOR410* interference. *TaCOR410i* represented the interference group. *GFPi* is the control of *TaCOR410i*. Scale bars = 12 cm. (**B**) Statistical analysis of the efficiency of *TaCOR410* knockdown by qRT-PCR assay. β-actin was used as the internal reference and the relative expression was calculated using the 2^−ΔΔCT^ method. (**C**) Expression of *TaCOR410* under dehydration treatment. β-actin was used as the internal reference and the relative expression was calculated using the 2^−ΔΔCT^ method. (**D**) The fresh weight was statistically analyzed after *TaCOR410* interference. All experiments were conducted in triplicate. The error is the average SD of three experimental data, and significant differences between the experimental and control groups were analyzed using the *t*-test (* *p* < 0.05; ** *p* < 0.01).

**Figure 6 plants-14-02726-f006:**
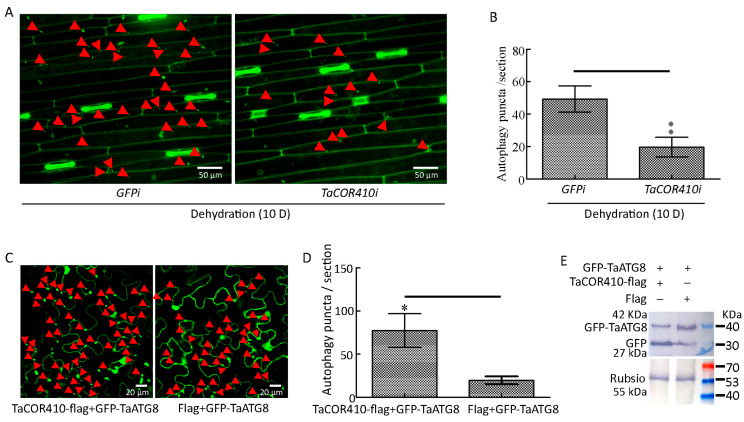
*TaCOR410* promotes autophagy in wheat. (**A**) TaATG8-labeled autophagy puncta was detected after *TaCOR410* interference under drought stress. *TaCOR410i* represented the interference group, *GFPi* is the control of *TaCOR410i*. Dehydration: 20% PEG-8000 was used to simulate drought stress. The red arrows indicate autophagy puncta. Scale bars = 50 μm. (**B**) Significance analysis of the number of autophagy puncta per section in Figure 6A. Three sections were used for analysis. (**C**) Autophagy puncta were indicated by the transfection of the *GFP-TaATG8* plasmid in tobacco leaves. Scale bars = 20 μm. Autophagy puncta were detected after co-transfection of *TaCOR410-flag*/*GFP-TaATG8* and *Flag*/*GFP-TaATG8* for 48 h. The red arrows indicate autophagy puncta. (**D**) Significance analysis of the number of GFP-TaATG8-labeled autophagy puncta per section in (**C**). Three sections were used for analysis. (**E**) Western blotting was used to detect the cleavage of GFP-TaATG8. Rubisco was used as the control. The SDS gel concentration was 12%. (* *p* < 0.05; ** *p* < 0.01).

**Figure 7 plants-14-02726-f007:**
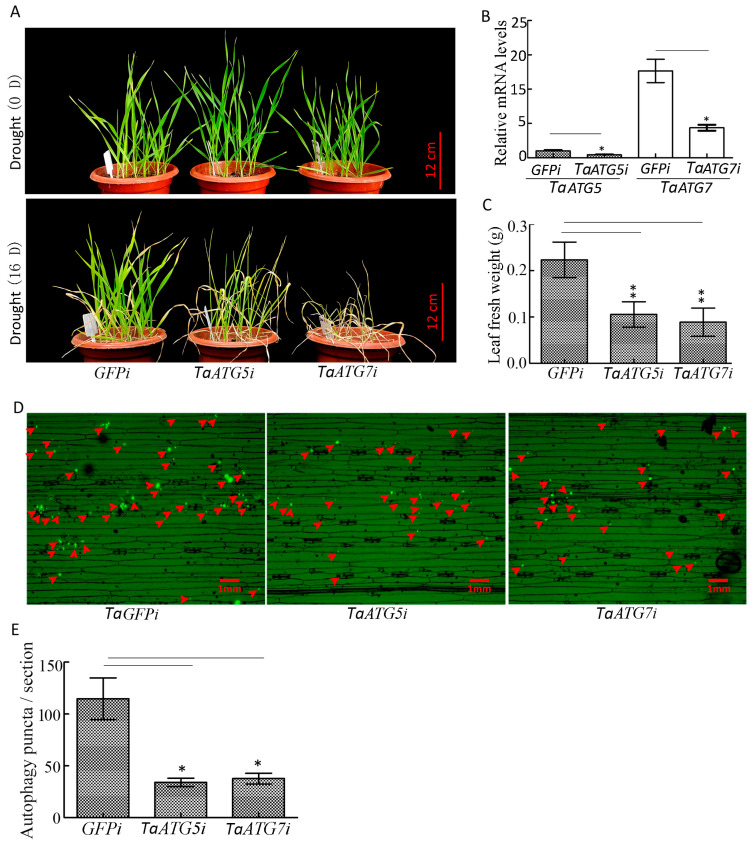
Interference with *TaATG5* or *TaATG7* reduces the drought resistance of wheat. (**A**) Phenotype of wheat seedlings after interference with *TaATG5* or *TaATG7* under drought stress. *TaATG5i* and *TaATG7i* represented the interference group, while *GFPi* was used as the control group. Scale bars = 12 cm. (**B**) qRT-PCR analysis of the knockdown efficiency of *TaATG5* and *TaATG7*. β-actin was used as the internal reference and the relative expression was calculated using the 2^−ΔΔCT^ method. (**C**) The fresh weight was statistically analyzed after interference with *TaATG5* or *TaATG7*. (**D**) Detection of autophagy puncta in wheat leaf after interference with *TaATG5* or *TaATG7* under drought stress. The red arrows indicate autophagy puncta. Scale bars = 1 mm. (**E**) Significance analysis of the number of autophagy puncta per section in (**D**). Three sections were used for analysis. The error is the average SD of three experiments, and significant differences between the experimental and control groups were analyzed using the *t*-test (* *p* < 0.05, ** *p* < 0.01).

**Figure 8 plants-14-02726-f008:**
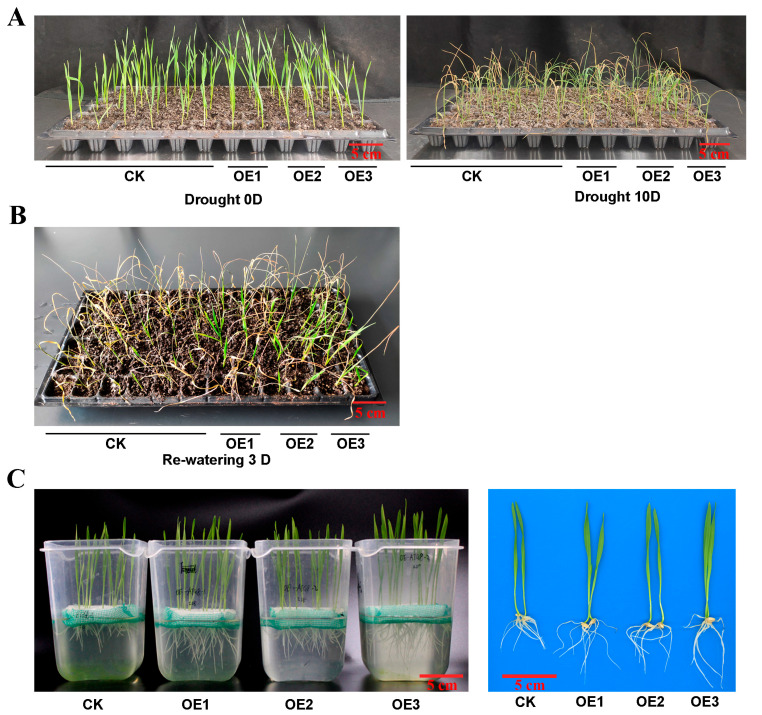
Overexpression of *TaATG8* improves the drought resistance of wheat. (**A**) Phenotype of *TaATG8* transgenic lines at the seedling stage under drought stress. CK: Fielder wheat; OE1, OE2, and OE3, three independent *TaATG8* transgenic lines. (**B**) Phenotype of seedlings after rehydration for 3 days. (**C**) Phenotypes of transgenic lines and controls were observed by treating the seedlings with 20% PEG-8000 to simulate drought. (**D**) The root and bud lengths were statistically compared between the transgenic lines and the control groups. (**E**) Transgenic detection of *TaATG8* in 1% agarose gel. (**F**) qRT-PCR analysis of the expression of *TaATG8* transgenic lines in wheat leaves. β-actin was used as the internal reference and the relative expression was calculated using the 2^−ΔΔCT^ method. All experiments were conducted in triplicate. Scale bars = 12 cm. The error is the average SD of three experiments, and significant differences between the experimental and control groups were analyzed using the *t*-test (** *p* < 0.01; *** *p* < 0.001).

**Figure 9 plants-14-02726-f009:**
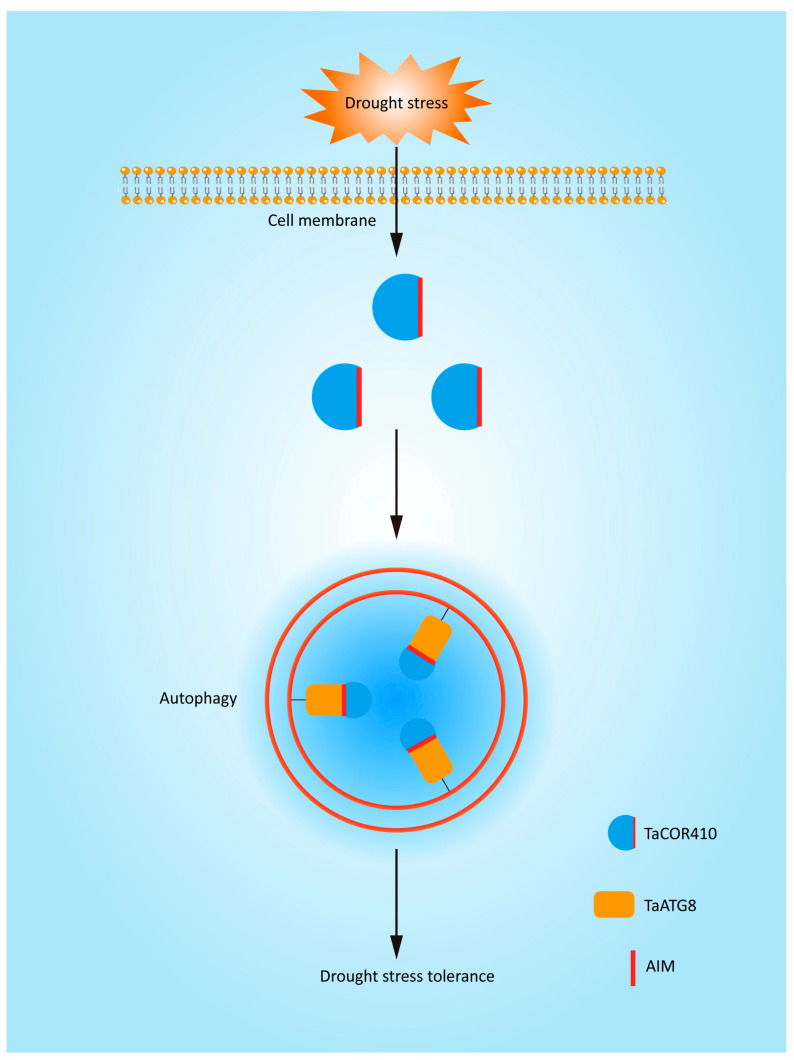
A model to explain the molecular mechanism of autophagy in response to drought stress. TaCOR410 interacts with TaATG8 through its AIM to promote autophagy under drought stress.

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
