# Peer review of "Dehydrin Protein TaCOR410 Improves Drought Resistance of Wheat Through Autophagy"

_plants, 2025, doi:10.3390/plants14172726_

Round 1
Reviewer 1 Report
Comments and Suggestions for Authors
The manuscript submitted by Yan et al. describes the role of wheat dehydrin TaCOR410 in drought responses. TaCOR410 confers resistance in an autophagy-dependent manner through the interaction with the ATG8 via a specific (AIM) motif. Interactions are verified by Co-IP as well as microscopy-based and pull-down assays. Interestingly, RNA interference with the expression of TaCOR410 compromises drought resistance in wheat seedlings and, in the same direction, ATG8 overexpression improves plant robustness during drought stress. This is an interesting hypothesis in the effort to improve our understanding of the mechanisms underlying stress responses in crops. However, in my opinion, the (scientific) writing is unclear and one is difficult to follow the statements and fully understand the rationale behind the experiments and the experimental setup in each case. The authors might consider asking for the opinion of a native English speaker and revising their manuscript extensively. Particular attention must be given to the materials and methods section, as critical information regarding the experiments shown is missing.
Some additional comments and suggestions are given below:
- The authors should add a few sentences at the beginning of each section of the results explaining the main point and giving sufficient background information for every experiment. This will improve clarity and readability of the manuscript (For instance, check critical information missing in lines 97-100).
- Autophagy puncta are used to monitor and quantify autophagy. However, in Fig. 1A and 7D, I am not convinced that the dots marked with the red arrowheads are indeed intracellular (Compare with dots shown in Fig. 6; Did the authors treat with cell-wall degrading enzymes after fixation?). When showing autophagy puncta, they should use the same settings and microscope. Along the same lines, what does puncta per view mean? The description provided in section 4.12 is not informative at all.
- The authors came across TaCOR410 using a library of ATG8 interaction proteins that was established using wheat ATG8 polyclonal antibody (Lines 114-115). This allowed the identification of TaCOR410 as an ATG8 interactor partner. However, surprisingly, there is no further information related to this experiment or the antibody employed for the construction of this library. Which antibody is this, and how was it produced? Did the author use this antibody in all the experiments? Please explain and provide any information and references missing related to the anti-ATG8 antibody (or antibodies) used.
- In general, materials and methods section and the figure legends lack critical information, hindering understanding.
- Please add the protein marker used in the materials and methods and the sizes of the protein bands next to the blot images. In addition, add the predicted sizes of the proteins used in Co-IPs or pull-down experiments in the figure legends and improve blot labeling.
- When referring wheat ATG8, use always TaATG8 instead of ATG8.
- Line 88, please revise the sentence.
- Line 143, please revise
-Line 419, Is this the correct agrobacterium strain?
- Be careful with the use of the word interfering.
- Many spelling errors can be found, and this is distracting.
Author Response
Reviewer1
The manuscript submitted by Yan et al. describes the role of wheat dehydrin TaCOR410 in drought responses. TaCOR410 confers resistance in an autophagy-dependent manner through the interaction with the ATG8 via a specific (AIM) motif. Interactions are verified by Co-IP as well as microscopy-based and pull-down assays. Interestingly, RNA interference of TaCOR410 compromises drought resistance in wheat seedlings and, in the same direction, ATG8 overexpression improves plant robustness during drought stress. This is an interesting hypothesis in the effort to improve our understanding of the mechanisms underlying stress responses in crops.
- However, in my opinion, the (scientific) writing is unclear and one is difficult to follow the statements and fully understand the rationale behind the experiments and the experimental setup in each case. The authors might consider asking for the opinion of a native English speaker and revising their manuscript extensively.
Answer: Regarding language issues, we have conducted comprehensive revisions with the support of professional editors at MDPI (https://www.mdpi.com/authors/english). Please refer to the revised manuscript for details.
- attention must be given to the materials and methods section, as critical information regarding the experiments shown is missing.
Answer: In the Methods and Materials section, we have incorporated additional key experimental details, as outlined below:
(1) In Section 4.2, We have incorporated a detailed method for generating TaATG8 transgenic materials, as described below:
“TaATG8 transgenic materials were obtained by Professor Li Genying’s group from the Crop Research Institute, Shandong Academy of Agricultural Sciences. Briefly, the open reading frame (ORF) of TaATG8 (TraesCS2A02G224000) was cloned into the binary vector pLGY02 at the SmaI and SpeI sites, forming a pLGY02-TaATG8 recombinant plasmid and maize ubiquitin promoter-driven target gene expression. The construct pLGY02-TaATG8 was used to transfsence of the transgene via PCR amplification, and Fielder was used as the negative control. Phenotypic identification of three group plants from each T4 line was performed at the seedling stage.”
- In Section 4.5, We have included the detailed preparation method of the TaATG8 polyclonal antibody, as described below:
“For the preparation of the TaATG8 polyclonal antibody, we followed the method outlined in a previous study[1]. Briefly, one Japanese white rabbit and one New Zealand white rabbit of experimental grade were raised. When the rabbits grew to 1-2 kg, 1 mL of well-mixed complete Freund's adjuvant and 1 mL of TaATG8 polypeptide (N-KTEHPLERRQAESARIRE-C) at a concentration of 0.7 mg/mL-1 were injected subcutaneously into each rabbit with a syringe. This stage was marked as day 1. On day 12, 1 mL of well-mixed incomplete Freund's adjuvant and 1 mL of polypeptide at a concentration of 0.35 mg mL-1 were injected subcutaneously into the rabbits. On days 26, 40, and 54, 1 mL of well-mixed incomplete Freund's adjuvant and 1 mL of polypeptide at a concentration of 0.35 mg mL-1 were injected intramuscularly into the rabbits. On day 66, the rabbit serum was taken for Western blot verification.”
- In Section 4.6, we have added the method for establishing the TaATG8 interacting protein library which is described in detail as follows:
“First, 1 mL of total protein of wheat leaf at a concentration of 2 mg mL-1 and 5 μL TaATG8 antibody at a concentration of 1.8 mg mL-1 were mixed and incubated on ice overnight; thereafter, 50 μL of Protein G Agarose (P2053, Beyotime, Shanghai, China) was added and incubated on ice for 3 hours; it was then centrifuged at 400×g for 5 minutes and the supernatant was discarded. The gel-like substance at the bottom of the tube was retained, washed 5 times with PBS, and 40 μL of PBS was added and mixed after homogenization. Next, 20 μL of 5× protein loading buffer (G2075, Servicebio, Wuhan, China) was added, boiled for 10 minutes, and then proceeded by means of SDS-PAGE electrophoresis, with subsequent staining with SDS gel. The gel was subjected to mass spectrometry identification and analysis after decolorization.
The samples were processed using a standard in-gel digestion protocol, with trypsin used for in-gel digestion of the gel slices [2]. The specific experimental steps were as follows: the stained gel was incubated with trypsin (Promega) overnight at 37℃. Peptides were extracted by means of incubation with 5% tallow fatty acid (TFA) for 1 h, followed by the addition of 2.5% TFA and 50% acetonitrile (ACN) at 37℃ for 1 h. The cleaved peptides were analyzed using a Nano LC-LTQ Orbitrap XL Liquid Chromatography–tandem mass spectrometry (LC-MS/MS) System (Thermo Scientific, Waltham, MA, USA). Data were analyzed using pFindStudio software (v.3.1.6)[3].”
- In Section 4.9, We have incorporated the sampling location of wheat leaves and provided detailed information regarding secondary antibodies for immunohistochemical procedures. These details are elaborated on as follows:
“Place the first leaf of the treated wheat seedling on the operating table, and gently make an incision in the middle part of the leaf with the tip of a dissecting knife. Then, using forceps, carefully peel off the lower epidermis from the leaf along the incision. The lower epidermis was treated with 4% paraformaldehyde at 4 ℃ overnight, placed in BSA and sealed at 37 ℃ for 1 h, incubated with a rabbit anti-ATG8 antibody diluted 1:200 in blocking buffer in TBS and incubated at 4 ℃ overnight, washed 3 times with PBS, incubated with the secondary antibody AF488-labeled Goat Anti-Rabbit IgG (A0423, Beyotime, Shanghai, China) diluted 1:1000 in BSA at 37 ℃ for 1 h......”
- In Section 4.11, we have added the specific vector construction process for the GFP cleavage assay, as well as the target proteins to be detected,which are described in detail as follows:
“The ORF of TaCOR410 (TraesCS6D02G234700) was cloned into the 35S-pBWA(V)KS-CCDB–flag vector at NdeI and EcoRI sites, forming a 35S-TaCOR410-Flag recombinant plasmid. GFP was linked with the N-terminal ORF of TaATG8 to generate the GFP-TaATG8 fragment; thereafter, the ligated fragment was cloned into the 35S-pBWA(V)HS-CCDB vector at the BsaI and Eco31I sites, forming a 35S-GFP-TaATG8 recombinant plasmid. The above plasmids, 35S-GFP-TaATG8, 35S-TaCOR410-flag, and 35S-flag, were transferred into Agrobacterium GV3101. The bacterial culture containing 35S-GFP-TaATG8 and the bacterial culture containing 35s-TaCOR410-flag or 35s-flag were evenly mixed at a ratio of 1:1 and centrifuged, and the supernatant was discarded; tobacco infection liquid was added to resuspend the bacteria and then injected into tobacco leaves, which were cultured for 48 h. The protein extract was analyszed with anti-GFP, and GFP-TaATG8 and free GFP were detected. In this assay, Rubisco was used as the reference.” (6) In Section 4.15, We have incorporated the quantitative software and significant difference analysis methods utilized in quantitative statistics, which are detailed as follows:
“Quantitative analysis of autophagy puncta using ImageJ software. The data are presented as the mean±SD. Significant differences between the two groups were determined using Student's t-test with GraphPad Prism 5 software. Asterisks indicate statistically significant differences (*, P < 0.05; **, P < 0.01).”
- Some additional comments and suggestions are given below:The authors should add a few sentences at the beginning of each section of the results explaining the main point and giving sufficient background information for every experiment. This will improve clarity andreadability of the manuscript (For instance, check critical information missing in lines 97-100).
Answer: We have incorporated brief introductory statements at the outset of each results section to clarify the primary objective and contextualize the experiments. This provides readers with a clear understanding of the rationale behind each study. The specifics are outlined below:
- Result 2.1:added “To investigate whether autophagy can respond to drought stress,ATG8-labeled autophagy puncta were used to detect plant tissue autophagy [4], and the expression of key autophagy genes was determined under dehydration conditions......”
- Result 2.2:added “To investigatethe molecular mechanism underlying the autophagy response to drought stress, a library of ATG8 interaction proteins was established by immunoprecipitating the total proteins of wheat leaves with a wheat ATG8 antibody and performing a mass spectrometry In this library, a wheat DHN protein, TaCOR410, was identified (Fig. 2A). Due to DHNs being intrinsically disordered proteins and showing a tendency to interact with other biomolecules, we investigated whether TaCOR410 functions as a cargo receptor in the autophagy pathway. In general, ATG8 can bind to its interactors through the ATG8 interaction motif (AIM; W/F/YX1X2L/I/V)[5,6]. Hence, we analyzed the sequence of TaCOR410 and the predicted 1 AIM motif (75-FSKL-78)......”
- Result 2.3:added “To further confirm the interaction of TaCOR410 with TaATG8, bimolecular fluorescence complementation (BiFC), co-immunoprecipitation (Co-IP), and pull-down experiments were conducted.......”
- Result 2.5:added “To examine whether TaCOR410 plays a rolein drought-induced autophagy, we investigated the effects of TaCOR410 on autophagy under drought stress......”
- Result 2.6:added “To elucidate the effects of autophagy on the drought resistance of wheat, the drought resistance phenotype of the seedlings was detected after autophagy was blocked by interference with TaATG5or TaATG7......”
- Result 2.7:added “To investigate the role of TaATG8under drought stress, we generated transgenic plants overexpressing TaATG8 (OE1, OE2, OE3), driven by the maize ubiquitin promoter, and Fielder was used as the receptor material.......”
- However, in Fig. 1A and 7D, I am not convinced that the dots marked with the red arrowheads are indeed intracellular (Compare with dots shown in Fig. 6;)
Answer:Firstly, ATG8-labeled autophagy puncta can be used as a common method of autophagy detection [4]; Secondly, “The results showed that the abundance of autophagy puncta increased significantly under dehydration for 3–24 h when compared with the control (H2O); with the increase in dehydration time, the greater the abundance of autophagy puncta (Fig. 1A, B)”; These results are consistent with the prevailing view that drought triggers plant cell autophagy. The most important is “Interference with TaATG5 or TaATG7 led to a significant reduction in autophagy puncta” and “ATG5 and ATG7 play key roles in autophagy, and inhibition of ATG5 or ATG7 represses the autophagy pathway [7]. Therefore, the fluorescent spots shown in Fig. 1A and 7D are indeed intracellular autophagy puncta.
- Did the authors treat with cell-wall degrading enzymes after fixation?
Answer: During the experiment, the cell wall did not need to be treated with cell-wall degrading enzymes after being fixed with paraformaldehyde, and the specific method in wheat is as follows: “Place the first leaf of the treated wheat seedling on the operating table, and gently make an incision in the middle part of the leaf with the tip of a dissecting knife. Then, using forceps, carefully peel off the lower epidermis from the leaf along the incision. The lower epidermis were treated with 4% paraformaldehyde at 4 ℃ overnight, placed in BSA and sealed at 37 ℃ for 1 h, incubated with a rabbit anti-ATG8 antibody diluted 1:200 in blocking buffer in TBS and incubated at 4 ℃ overnight, washed 3 times with PBS, incubated with the secondary antibody AF488-labeled Goat Anti-Rabbit IgG (A0423, Beyotime, Shanghai, China) diluted 1:1000 in BSA at 37 ℃ for 1 h, and washed 3 times with PBS. Thereafter, the lower epidermis tissue of the wheat leaves was placed on a glass slide for tableting, and a fluorescence microscope (HT7700; Hitachi, Tokyo, Japan) and a confocal microscope (FV3000; Olympus, Japan) were used to observe the samples and obtain images.”
- When showing autophagy puncta, they should use the same settings and microscope. Along the same lines, what does puncta per view mean?
Answer: we have changed it to “Significance analysis of the number of autophagy puncta per section in Figs(1A,6A,6C,7A,7D). Three sections were used for analysis”.
7、The description provided in section 4.12 is not informative at all.
Answer: We have made some modifications, as described in section 4.15 “Quantitative analysis of autophagy puncta using ImageJ software. The data are presented as the mean±SD. Significant differences between the two groups were determined using Student's t-test with GraphPad Prism 5 software. Asterisks indicate statistically significant differences (*, P < 0.05; **, P < 0.01).”
- The authors came across TaCOR410 using a library of ATG8 interaction proteins that was established using wheat ATG8 polyclonal antibody (Lines 114-115). This allowed the identification of TaCOR410 as an ATG8 interactor partner. However, surprisingly, there is no further information related to this experiment or the antibody employed for the construction of this library. Which antibody is this, and how was it produced? Did the author use this antibody in all the experiments? Please explain and provide any information and references missing related to the anti-ATG8 antibody (or antibodies) used.
Answer: In the manuscript, the TaATG8 antibody is employed for the detection of autophagy puncta and the establishment of a TaATG8 interacting protein library. The development process of this antibody is as follows:
“For the preparation of the TaATG8 polyclonal antibody, we followed the method outlined in a previous study[1]. Briefly, one Japanese white rabbit and one New Zealand white rabbit of experimental grade were raised. When the rabbits grew to 1-2 kg, 1 mL of well-mixed complete Freund's adjuvant and 1 mL of TaATG8 polypeptide (N-KTEHPLERRQAESARIRE-C) at a concentration of 0.7 mg/mL were injected subcutaneously into each rabbit with a syringe. This stage was marked as day 1. On day 12, 1 mL of well-mixed incomplete Freund's adjuvant and 1 mL of polypeptide at a concentration of 0.35 mg mL-1 were injected subcutaneously into the rabbits. On days 26, 40, and 54, 1 mL of well-mixed incomplete Freund's adjuvant and 1 mL of polypeptide at a concentration of 0.35 mg mL-1 were injected intramuscularly into the rabbits. On day 66, the rabbit serum was taken for Western blot verification.”
We also employed this antibody to establish a library of TaATG8 interacting proteins; the detailed procedure is outlined as follows: “First, 1 mL of total protein of wheat leaf at a concentration of 2 mg mL-1 and 5 μL TaATG8 antibody at a concentration of 1.8 mg/mL were mixed and incubated on ice overnight; thereafter, 50 μL of Protein G Agarose (P2053, Beyotime, Shanghai, China) was added and incubated on ice for 3 hours; it was then centrifuged at 400×g for 5 minutes and the supernatant was discarded. The gel-like substance at the bottom of the tube was retained, washed 5 times with PBS, and 40 μL of PBS was added and mixed after homogenization. Next, 20 μL of 5× protein loading buffer (G2075, Servicebio, Wuhan, China) was added, boiled for 10 minutes, and then proceeded by means of SDS-PAGE electrophoresis, with subsequent staining with SDS gel. The gel was subjected to mass spectrometry identification and analysis after decolorization.
The samples were processed using a standard in-gel digestion protocol, with trypsin used for in-gel digestion of the gel slices [2]. The specific experimental steps were as follows: the stained gel was incubated with trypsin (Promega) overnight at 37℃. Peptides were extracted by means of incubation with 5% tallow fatty acid (TFA) for 1 h, followed by the addition of 2.5% TFA and 50% acetonitrile (ACN) at 37℃ for 1 h. The cleaved peptides were analyzed using a Nano LC-LTQ Orbitrap XL Liquid Chromatography–tandem mass spectrometry (LC-MS/MS) System (Thermo Scientific, Waltham, MA, USA). Data were analyzed using pFindStudio software (v.3.1.6)[3].”
- In general, materials and methods section and the figure legends lack critical information, hindering understanding.
Answer: We have supplemented the critical information as answered in Question 2.
- Please add the protein marker used in the materials and methods and the sizes of the protein bands next to the blot images. In addition, add the predicted sizes of the proteins used in Co-IPs or pull-down experiments in the figure legends and improve blot labeling.
Answer: We have added the protein marker used in the Materials and Methods, the sizes of the protein bands next to the blot images and the predicted sizes of the proteins used in Co-IP or pull-down experiments in the figure legends. We have also improved blot labeling. Please refer to the revised version.
11.When referring wheat ATG8, use always TaATG8 instead of ATG8.
Answer: We have used TaATG8 throughout the manuscript.
- Line 88, please revise the sentence.
Answer:We have revised it to “The interaction of rice YSK2-type dehydrin (OsDhn-Rab16D) with a prolyl cis-trans isomerase (OsFKBP, Os02g52290) improved drought resistance through ABA signaling [8].”
- Line 143, please revise
Answer: We have revised it to “BiFC assay confirming the interaction of TaCOR410 with TaATG8 in vivo”
- Line 419, Is this the correct agrobacterium strain?
Answer: GV3101 is correct; we have corrected throughout the manuscript.
- Be careful with the use of the word interfering.
Answer:We used “interferrence” instead of “interfering”
- Many spelling errors can be found, and this is distracting.
Answer:We have corrected throughout the manuscript.

Reviewer 2 Report
Comments and Suggestions for Authors
I go through the manuscript plants-3819609, which tried to investigates how the TaCOR410 protein in wheat improves drought resistance by interacting with the ATG8 protein to boost autophagy (a cellular "cleanup" process). The main goal is to understand if this interaction helps wheat survive dry conditions. However this hypothesis is wrongly intercepted. First, this paper has serious flaws in its methods. The experiments aren't consistent or clear details like how samples were prepared are vague. The authors don’t use strong statistics to back their claims, making the results shaky. It feels like they threw together data without proper validation. The study offers no real-world use such as even if true, it doesn’t show how this could help farmers grow better crops or fight hunger.
Second, and the most importantly the manuscript drafted in non scientific language, lack originality and work is poorly presented. It recycles old ideas about drought and autophagy without fresh insights like a dump of lab data with zero innovation. The writing is unscientific, packed with typos and grammar errors, making it hard to trust. While the core idea (protein interactions) has some science merit, the messy methods and amateurish language fall far below standards for journals like Plants. That’s why such research stalls progress, and this journal struggles to grow. I must recommend rejection.
Comments on the Quality of English Languagemost importantly the manuscript drafted in non scientific language, lack originality and work is poorly presented. The writing is unscientific, packed with typos and grammar errors, making it hard to trust.
Author Response
Review 2
- I go through the manuscript plants-3819609, which tried to investigates how the TaCOR410 protein in wheat improves drought resistance by interacting with the ATG8 protein to boost autophagy (a cellular "cleanup" process). The main goal is to understand if this interaction helps wheat survive dry conditions. However this hypothesis is wrongly intercepted. First, this paper has serious flaws in its methods. The experiments aren't consistent or clear details like how samples were prepared are vague. The authors don’t use strong statistics to back their claims, making the results shaky. It feels like they threw together data without proper validation. The study offers no real-world use such as even if true, it doesn’t show how this could help farmers grow better crops or fight hunger.
- Second, and the most importantly the manuscript drafted in non scientific language, lack originality and work is poorly presented. It recycles old ideas about drought and autophagy without fresh insights like a dump of lab data with zero innovation. The writing is unscientific, packed with typos and grammar errors, making it hard to trust. While the core idea (protein interactions) has some science merit, the messy methods and amateurish language fall far below standards for journals like Plants. That’s why such research stalls progress, and this journal struggles to grow. I must recommend rejection.
We have summarized this reviewer's comments. The primary concerns raised by the reviewer are as follows:The language used in the manuscript is largely non-scientific, which may impede clear communication of the research findings. Additionally, the Materials and Methods section, along with the figure and table legends, lacks essential information that is crucial for understanding the study's methodology and results. These shortcomings have led to a negative evaluation from the reviewer.
- Regarding language issues, we have conducted comprehensive revisions with the support of professional editors at MDPI (https://www.mdpi.com/authors/english). Please refer to the revised manuscript for details.
- In the Methods and Materials section, we have incorporated additional key experimental details, as outlined below:
(1) In Section 4.2, We have incorporated a detailed method for generating TaATG8 transgenic materials, as described below:
“TaATG8 transgenic materials were obtained by Professor Li Genying’s group from the Crop Research Institute, Shandong Academy of Agricultural Sciences. Briefly, the open reading frame (ORF) of TaATG8 (TraesCS2A02G224000) was cloned into the binary vector pLGY02 at the SmaI and SpeI sites, forming a pLGY02-TaATG8 recombinant plasmid and maize Ubiquitin promoter-driven target gene expression. The construct pLGY02-TaATG8 was used to transform the drought-sensitive variety Fielder through Agrobacterium tumefaciens (Agrobacterium)-mediated transformation [1]. T1-T4 plants were tested for the presence of the transgene via PCR amplification, and Fielder was used as the negative control. Phenotypic identification of three group plants from each T4 line was performed at the seedling stage.”
- In Section 4.5, We have included the detailed preparation method of the TaATG8 polyclonal antibody, as described below:
“For the preparation of the TaATG8 polyclonal antibody, we followed the method outlined in a previous study[2]. Briefly, one Japanese white rabbit and one New Zealand white rabbit of experimental grade were raised. When the rabbits grew to 1-2 kg, 1 mL of well-mixed complete Freund's adjuvant and 1 mL of TaATG8 polypeptide (N-KTEHPLERRQAESARIRE-C) at a concentration of 0.7 mg mL-1 were injected subcutaneously into each rabbit with a syringe. This stage was marked as day 1. On day 12, 1 mL of well-mixed incomplete Freund's adjuvant and 1 mL of polypeptide at a concentration of 0.35 mg mL-1 were injected subcutaneously into the rabbits. On days 26, 40, and 54, 1 mL of well-mixed incomplete Freund's adjuvant and 1 mL of polypeptide at a concentration of 0.35 mg mL-1 were injected intramuscularly into the rabbits. On day 66, the rabbit serum was taken for Western blot verification.”
- In Section 4.6, we have added the method for establishing the TaATG8 interacting protein library which is described in detail as follows:
“First, 1 mL of total protein of wheat leaf at a concentration of 2 mg mL-1 and 5 μL TaATG8 antibody at a concentration of 1.8 mg/mL were mixed and incubated on ice overnight; thereafter, 50 μL of Protein G Agarose (P2053, Beyotime, Shanghai, China) was added and incubated on ice for 3 hours; it was then centrifuged at 400×g for 5 minutes and the supernatant was discarded. The gel-like substance at the bottom of the tube was retained, washed 5 times with PBS, and 40 μL of PBS was added and mixed after homogenization. Next, 20 μL of 5× protein loading buffer (G2075, Servicebio, Wuhan, China) was added, boiled for 10 minutes, and then proceeded by means of SDS-PAGE electrophoresis, with subsequent staining with SDS gel. The gel was subjected to mass spectrometry identification and analysis after decolorization.
The samples were processed using a standard in-gel digestion protocol, with trypsin used for in-gel digestion of the gel slices [3]. The specific experimental steps were as follows: the stained gel was incubated with trypsin (Promega) overnight at 37℃. Peptides were extracted by means of incubation with 5% tallow fatty acid (TFA) for 1 h, followed by the addition of 2.5% TFA and 50% acetonitrile (ACN) at 37℃ for 1 h. The cleaved peptides were analyzed using a Nano LC-LTQ Orbitrap XL Liquid Chromatography–tandem mass spectrometry (LC-MS/MS) System (Thermo Scientific, Waltham, MA, USA). Data were analyzed using pFindStudio software (v.3.1.6)[4].
- In Section 4.9, We have incorporated the sampling location of wheat leaves and provided detailed information regarding secondary antibodies for immunohistochemical procedures. These details are elaborated on as follows:
“Place the first leaf of the treated wheat seedling on the operating table, and gently make an incision in the middle part of the leaf with the tip of a dissecting knife. Then, using forceps, carefully peel off the lower epidermis from the leaf along the incision. The lower epidermis was treated with 4% paraformaldehyde at 4 ℃ overnight, placed in BSA and sealed at 37 ℃ for 1 h, incubated with a rabbit anti-ATG8 antibody diluted 1:200 in blocking buffer in TBS and incubated at 4 ℃ overnight, washed 3 times with PBS, incubated with the secondary antibody AF488-labeled Goat Anti-Rabbit IgG (A0423, Beyotime, Shanghai, China) diluted 1:1000 in BSA at 37 ℃ for 1 h, and washed 3 times with PBS. Thereafter, the lower epidermis tissue of the wheat leaves was placed on a glass slide for tableting, and a fluorescence microscope (HT7700; Hitachi, Tokyo, Japan) and a confocal microscope (FV3000; Olympus, Japan) were used to observe the samples and obtain images.”
- In Section 4.11, we have added the specific vector construction process for the GFP cleavage assay, as well as the target proteins to be detected; which aredescribed in detail as follows.
“The ORF of TaCOR410 (TraesCS6D02G234700) was cloned into the 35S-pBWA(V)KS-CCDB–flag vector at NdeI and EcoRI sites, forming a 35S 35S-TaCOR410-Flag recombinant plasmid. GFP was linked with the N-terminal ORF of TaATG8 to generate the GFP-TaATG8 fragment; thereafter, the ligated fragment was cloned into the 35S-pBWA(V)HS-CCDB vector at the BsaI and Eco31I sites, forming a 35S-GFP-TaATG8 recombinant plasmid. The above plasmids, 35S-GFP-TaATG8, 35S-TaCOR410-flag, and 35S-flag, were transferred into Agrobacterium GV3101. The bacterial culture containing 35S-GFP-TaATG8 and the bacterial culture containing 35S-TaCOR410-flag or 35S-flag were evenly mixed at a ratio of 1:1 and centrifuged, and the supernatant was discarded; tobacco infection liquid was added to resuspend the bacteria and then injected into tobacco leaves, which were cultured for 48 h. The protein extract was analyzed with anti-GFP, and GFP-TaATG8 and free GFP were detected. In this assay, Rubisco was used as the reference.”
- In Section 4.15, We have incorporated the quantitative software and significant difference analysis methods utilized in quantitative statistics, which are detailed as follows:
“Quantitative analysis of autophagy puncta using ImageJ software. The data are presented as the mean±SD. Significant differences between the two groups were determined using Student's t-test with GraphPad Prism 5 software. Asterisks indicate statistically significant differences (*, P < 0.05; **, P < 0.01).”
- Moreover, we have summarized that the innovation of this manuscript is mainly reflected in the following aspects:
- In wheat,TaCOR410 can improve drought resistance through autophagy.
- TaCOR410 interacts with TaATG8 to promote autophagy.
- TaCOR410 interacts with TaATG8 at the AIM motif.
- Moreover, we have refined the depiction of the interplay between TaCOR410 and TaATG8; the specifics are outlined as follows:
“To further confirm the interaction of TaCOR410 with TaATG8, bimolecular fluorescence complementation (BiFC), co-immunoprecipitation (Co-IP), and pull-down experiments were conducted. The BiFC results showed that the yellow fluorescence signal was observed only when TaATG8-nYFP and TaCOR410-cYFP were co-transfected into the tobacco leaves and wheat protoplasts; in comparison, no yellow fluorescence was detected in the negative controls. However, the yellow fluorescence signal disappeared when the AIM motif FSKL (TaCOR410-cYFP) was mutated into ASKA (TaCOR410ASKA-cYFP) (Fig. 3A–C).
In addition, Co-IP was also performed to verify the results. TaCOR410-flag/GFP-TaATG8 or TaCOR410ASKA-flag/GFP-TaATG8 were co-expressed in tobacco leaves. Protein was immunoprecipitated with anti-GFP beads and detected with anti-flag. Only TaCOR410-flag and GFP-TaATG8 were able to bind specifically (Fig. 4A). Lastly, we performed pull-down assays to confirm this interaction using GFP, TaCOR410-GFP, or TaCOR410ASKA-GFP protein as bait to precipitate the His-TaATG8 protein. As shown in Fig. 4B, the TaCOR410-GFP protein bound specifically to His-TaATG8; however, neither GFP nor TaCOR410ASKA-GFP exhibited the same behavior. From the above results, it can be concluded that wheat DHN protein TaCOR410 interacts with TaATG8 in the AIM motif both in vitro and in vivo.”
- Zhang, S.; Zhang, R.; Song, G.; Gao, J.; Li, W.; Han, X.; Chen, M.; Li, Y.; Li, G. Targeted mutagenesis using the Agrobacterium tumefaciens-mediated CRISPR-Cas9 system in common wheat. BMC plant biology 2018, 18, 302, doi:10.1186/s12870-018-1496-x.
- LI Yong-Bo, C.D.-Z., HUANG Chen, SUI Xin-Xia, FAN Qing-Qi and CHU Xiu-Sheng. Preparation of highly specific wheat ATG8 antibody and its application in the detection of autophagy. ACTA AGRONOMICA SINICA 2022, 48(9), 2390-2399, doi:10.3724/SP.J.1006.2022.11070.
- Rappsilber, J.; Mann, M.; Ishihama, Y. Protocol for micro-purification, enrichment, pre-fractionation and storage of peptides for proteomics using StageTips. Nat Protoc 2007, 2, 1896-1906, doi:10.1038/nprot.2007.261.
- Chi, H.; Liu, C.; Yang, H.; Zeng, W.F.; Wu, L.; Zhou, W.J.; Wang, R.M.; Niu, X.N.; Ding, Y.H.; Zhang, Y.; et al. Comprehensive identification of peptides in tandem mass spectra using an efficient open search engine. Nat Biotechnol 2018, doi:10.1038/nbt.4236.

Round 2
Reviewer 1 Report
Comments and Suggestions for Authors
The authors have addressed most of my concerns and the revised manuscript is improved.
Only a few suggestions:
The authors might consider redrawing the model presented in figure 9.
As is, it is very poor and not entirely clear what is their point,
what do they want to stress out (cellular, organism level, relative sizes of proteins motifs, cell boundaries etc).
Moreover, they should provide a clear description of what is shown in the figure legend.
Fig. 6 legend: it is an anti- ..rabbit antibody..
Author Response
Dear editor
First and foremost, we would like to convey our profound appreciation to the editor for affording us the opportunity to enhance our manuscript. In response to the reviewers' insightful comments, we have conducted meticulous, point-by-point revisions, with the amended portions highlighted in red for clarity.
The authors have addressed most of my concerns and the revised manuscript is improved.
Only a few suggestions:
The authors might consider redrawing the model presented in figure 9.
As is, it is very poor and not entirely clear what is their point,
what do they want to stress out (cellular, organism level, relative sizes of proteins motifs, cell boundaries etc).
Answer: We appreciate your valuable suggestion. In accordance with your feedback, we have meticulously redrawn Figure 9, which is now presented in the revised manuscript (please refer to Fig. 9)..
Moreover, they should provide a clear description of what is shown in the figure legend.
Fig. 6 legend: it is an anti- ..rabbit antibody..
Answer: We acknowledge your valuable suggestion. In response, we have incorporated comprehensive explanations for Figs. 1-8, which are now detailed in the figure legends of the revised manuscript. Please refer to the updated legends for Figs. 1-8 for further information.

Reviewer 2 Report
Comments and Suggestions for Authors
The authors addressed al the comments I raised successfully and I agree for the publication of this manuscript.
Author Response
The authors addressed al the comments I raised successfully and I agree for the publication of this manuscript.
Thank you very much